# Cyclic Hypoxia Conditioning Alters the Content of Myoblast-Derived Extracellular Vesicles and Enhances Their Cell-Protective Functions

**DOI:** 10.3390/biomedicines9091211

**Published:** 2021-09-13

**Authors:** Yan Yan, Tingting Gu, Stine Duelund Kaas Christensen, Junyi Su, Thomas Ravn Lassen, Marie Vognstoft Hjortbak, IJu Lo, Susanne Trillingsgaard Venø, Andrea Erzsebet Tóth, Ping Song, Morten Schallburg Nielsen, Hans Erik Bøtker, Blagoy Blagoev, Kim Ryun Drasbek, Jørgen Kjems

**Affiliations:** 1Interdisciplinary Nanoscience Center, Aarhus University, 8000 Aarhus, Denmark; yanyan@inano.au.dk (Y.Y.); junyi@inano.au.dk (J.S.); ijulo@inano.au.dk (I.L.); psong@inano.au.dk (P.S.); 2Omiics ApS, 8200 Aarhus, Denmark; susanne.veno@omiics.com; 3Center of Functionally Integrative Neuroscience, Department of Clinical Medicine, Aarhus University, 8000 Aarhus, Denmark; gutt@cfin.au.dk (T.G.); ryun@cfin.au.dk (K.R.D.); 4Department of Biochemistry and Molecular Biology, University of Southern Denmark, 5230 Odense, Denmark; sdkc@bmb.sdu.dk (S.D.K.C.); bab@bmb.sdu.dk (B.B.); 5Department of Cardiology, Aarhus University Hospital, Skejby, 8200 Aarhus, Denmark; thomasravnl@clin.au.dk (T.R.L.); hjortbak@clin.au.dk (M.V.H.); heb@dadlnet.dk (H.E.B.); 6Department of Biomedicine, Aarhus University, 8000 Aarhus, Denmark; toth@biomed.au.dk (A.E.T.); mn@biomed.au.dk (M.S.N.); 7Department of Molecular Biology and Genetics, Aarhus University, 8000 Aarhus, Denmark

**Keywords:** remote ischemic conditioning, myoblast, cyclic hypoxia-reoxygenation, extracellular vesicles, microRNAs, proteins

## Abstract

Remote ischemic conditioning (RIC) is a procedure that can attenuate ischemic-reperfusion injury by conducting brief cycles of ischemia and reperfusion in the arm or leg. Extracellular vesicles (EVs) circulating in the bloodstream can release their content into recipient cells to confer protective function on ischemia-reperfusion injured (IRI) organs. Skeletal muscle cells are potential candidates to release EVs as a protective signal during RIC. In this study, we used C2C12 cells as a model system and performed cyclic hypoxia-reoxygenation (HR) to mimic RIC. EVs were collected and subjected to small RNA profiling and proteomics. HR induced a distinct shift in the miRNA profile and protein content in EVs. HR EV treatment restored cell viability, dampened inflammation, and enhanced tube formation in in vitro assays. In vivo, HR EVs showed increased accumulation in the ischemic brain compared to EVs secreted from normoxic culture (N EVs) in a mouse undergoing transient middle cerebral artery occlusion (tMCAO). We conclude that HR conditioning changes the miRNA and protein profile in EVs released by C2C12 cells and enhances the protective signal in the EVs to recipient cells in vitro.

## 1. Introduction

Remote ischemic conditioning (RIC) is a therapeutic procedure that attenuates ischemia-reperfusion injury (IRI) by repeated temporary occlusion and release of blood flow to an effector organ distant to the target organ exposed to ischemia and reperfusion. In clinical practice, the effector organ is typically the upper limb since repetitive occlusion of blood flow and reperfusion is easily achieved with an ordinary blood pressure cuff [1]. However, the nature of the protective signals and their transmission from the remote conditioned organ to the damaged tissue remain unclear. Three overall pathways have been suggested: (1) a humoral pathway [2,3,4,5,6], (2) a neuronal pathway [4,6,7], and (3) an immunological pathway [6,8]. Candidate mediators of RIC protection include bradykinin [9], stromal-derived factor 1-alpha (SDF1-a) [10], adenosine [11], and extracellular vesicles [12,13].

Extracellular vesicles (EVs) are membrane-encapsulated vesicles released from cells. Exosomes are a subtype of EVs formed inside the multivesicular endosome (MVE) and secreted upon fusion of MVEs with the plasma membrane [14]. Proteins enriched in exosomes include tetraspannins (CD9, CD63, and CD81), heat-shock proteins, and TSG101 [15,16]. EVs have been found to act in cell–cell communication by delivering active biomolecules, such as RNA, proteins, and lipids to the receiver cells [14,17,18,19]. Different mechanisms of content packaging, recipient cell targeting, and signal transmission of EVs have been investigated [19]. Of the many molecules within EVs, microRNAs (miRNAs) are of special interest as these non-coding RNAs are ~22 nucleotides (nts) in size and regulate gene expression by interacting with mRNAs [20,21]. Previous studies reported that miRNAs can be encapsulated in EVs and circulate in biofluids, which may lead to the delivery of miRNAs to recipient cells [22,23,24]. After EVs are released from their donor cells, the integrins and receptors on EVs may target them to specific recipient cells where they deliver their protein and RNA cargo via endocytosis or fusion with the plasma membrane [14,19].

A number of studies have investigated the protective function of EVs during IRI. Vicencio et al. demonstrated that heat shock protein 70 (Hsp70) on plasma exosomes collected from RIC-treated rats and human patients, activated the toll-like receptor 4 (TLR4) pathway in cardiomyocytes leading to phosphorylation of heat shock protein 27 (Hsp27) to achieve cardioprotection [25]. Recently, Davidson et al. showed that exosomes collected from endothelial cell culture reduced IRI-induced cardiomyocyte death by activating the ERK1/2 MAPK signaling pathway [26]. Li et al. showed that RIC significantly increased miR-144 release in mouse myocardium and in the plasma of humans and mice [27]. Moreover, the study also showed that an increased level of miR-144 in plasma was found in the EV-poor fraction, but not in EVs, while the miR-144 precursor in plasma EVs showed a four-fold increase after RIC [27]. In another study, Wen et al. demonstrated that miR-24 was enriched in EVs purified from plasma from RIC-treated rats, and this miRNA showed a function in reducing oxidative stress-mediated injury and apoptosis of recipient cells [23]. The RIC procedure exerts pressure on the muscles, and skeletal muscle cells could therefore be responsible for the release of specific EVs as a protective signal during RIC. In our previous study, we found that blood flow restricted exercise (BFRE) on humans, which also creates ischemic conditions, altered the miRNA profile in EVs purified from human plasma and that BFRE-EVs induced muscle stem cell (MuSC, satellite cell) activation and proliferation [28]. During BFRE, muscle contractions from exercise create stress on the muscle cells [28], and muscle cells could be one cell type that secretes BFRE EVs, making muscle cells a very attractive cell type to study. Indeed, EVs secreted from myoblasts in culture have been shown to promote pre-osteoblast differentiation to osteoblasts by transferring miR-27a-3p to down-regulate the expression of adenomatous polyposis coli (APC) [29]. However, little work has been carried out to study the role of EVs released from RIC-treated myoblasts. Cai et al. found that RIC increased the level of IL-10 in the plasma of mice and that the increase in IL-10 depended on the expression of hypoxia-inducible factor (HIF) 1α [30]. They used C2C12 cells as a model and introduced cyclic hypoxia-reoxygenation to mimic RIC to validate the correlation of *IL-10* expression and *HIF-1α* expression [30]. Here, we cultured C2C12 cells under modified cyclic hypoxia-reoxygenation (HR) as a mimic for RIC and found that this treatment altered the expression of miRNAs and proteins in the released EVs. Moreover, compared to N EVs, HR EVs had stronger positive effects on cell viability, inflammation, and angiogenesis in cell culture and showed increased accumulation in the ischemic hemisphere in a transient middle cerebral artery occlusion (tMCAO) mouse model. Thus, our study suggests that myoblast EVs secreted under HR treatment can have a protective function in recipient cells, thereby indicating that myoblast EVs secreted under RIC may be the carrier of the protective signal.

## 2. Materials and Methods

### 2.1. Cell Culture and Cyclic Hypoxia-Reoxygenation (HR) Treatment

The cyclic hypoxia-reoxygenation (HR) procedure was performed by culturing cells with 1% O_2_ for 10 min in a hypoxic chamber followed by 20% O_2_ for 10 min in a normoxic incubator for four cycles (temperature and CO_2_ levels were kept constant). The number of cycles of hypoxia and normoxia was chosen based on previous studies by Bøtker et al. and Johnsen et al. [31,32]. The 10 min duration was chosen from Cai et al. [30]. In order to equilibrate oxygen levels rapidly, the hypoxic chamber was cleaned using 75% ethanol and the cell culture dish lid was removed while in the hypoxic chamber. The dishes were covered when they were transferred to normoxic conditions.

The mouse myoblast cell line C2C12 was purchased from ATCC (Manassas, VA, USA) and cultured in Dulbecco’s modified Eagle’s medium (DMEM) (ThermoFisher, Waltham, MA, USA) supplemented with 10% fetal bovine serum (FBS) (ThermoFisher, Waltham, MA, USA) and 1% penicillin-streptomycin (PS) (ThermoFisher, Waltham, MA, USA) at 37 °C with 5% CO_2_/95% air. The media was replaced every 2–3 days until the cells reached 90% confluency. Then, cells were subcultured and seeded in 100 mm dishes with 10 mL culture media. To harvest EVs, the cells were washed twice using 10 mL of phosphate-buffered saline (PBS), and the media was replaced by 5 mL of EV collection media when the cell confluency reached 100%. The EV collection media was prepared by adding 10% exosome-depleted FBS (ThermoFisher, Waltham, MA, USA) and 1% PS to DMEM. Half the dishes were cultured at 37 °C with 5% CO_2_ and 95% ambient air (normoxic conditions, N), while the other half were subjected to HR. After the HR treatment, 5 mL of additional EV collection media was added to both HR and N cells cultures, and the cells were cultured at normoxic conditions for 24 h before EV collection.

### 2.2. QRT-PCR for Cells

Cellular RNA was purified using Trizol (Invitrogen, Waltham, MA, USA) according to the manufacturer’s protocol. RNA concentration and quality were determined using a Nanodrop device (ThermoFisher, Waltham, MA, USA). One microgram of total cellular RNA was used for cDNA synthesis using the RevertAid RT Kit (Thermo Fisher Scientific, Waltham, MA, USA). qPCR of *IL-1β*, *TNF-α*, *GADPH*, and *HIF-1α* was performed using the primers listed in Appendix A (Sigma-Aldrich, St. Louis, MO, USA) and detected with aLightCycler 480 SYBR Green I Master (Roche, Basel, Switzerland) on a LightCycler 480 Instrument II (Roche, Basel, Switzerland). Three batches of cells were included and each sample was analyzed in triplicates. *GADPH* was used to normalize the data.

### 2.3. EV Isolation

EVs secreted from C2C12 cells were isolated using serial centrifugations from culture media [33]. The collected media was centrifuged at 300× *g* for 10 min at 4 °C and the supernatant was re-centrifuged at 2000× *g* for 10 min at 4 °C. The supernatant was centrifuged at 15,500× *g* for 30 min at 4 °C and the collected supernatant was passed through a 0.22 μm ACET/Prefilter (Sigma-Aldrich, St. Louis, MO, USA). The EVs were pelleted by ultracentrifugation at 100,000× *g* for 2 h at 4 °C (Beckman-Coulter (Brea, CA, USA), Optima L-80-XP ultracentrifuge, type 60Ti rotor) and re-suspended in 200 μL PBS. The EVs isolated from normoxic conditions are referred to as N EV, and the EVs isolated from HR conditions are called HR EV.

### 2.4. Nanoparticle Tracking Analysis (NTA) and Protein Measurements

The purified EVs were diluted at 1:5000 using PBS and analyzed using a NanoSight LM10 (Malvern Instruments Ltd., Malvern, UK) with a 405 nm laser. The settings used for data acquisition are as follows: measurements were performed five times with 60 s video capture of each sample with the camera level set at 15 and detection threshold at 10 for all analyses. The data was analyzed using NTA software version 3.1 (Nanosight, Amesbury, UK) to determine the sample concentration and EV size.

The protein concentration of re-suspended EVs was measured using a Qubit Protein Assay Kit according to the manufacturer’s protocol (Thermo Fisher Scientific, Waltham, MA, USA).

### 2.5. Transmission Electron Microscopy (TEM)

EV morphology was determined using a negative stain by Tecnai G2 Spirit TEM (Field Electron and Ion Company, FEI, Hillsboro, OR, USA). Briefly, EVs samples were applied on a carbon film-coated 400 mesh copper grid (microscopy products for Science and Industry) and stained with uranyl formate. The film was left to dry for 5–10 min and transferred into the electron microscope at room temperature. Images were acquired at a magnification of 21,000× (0.50 nm/pixel).

### 2.6. Western Blot

Cells were lysed with RIPA buffer (Invitrogen, Waltham, MA, USA) containing protease inhibitors (Complete Ultra Tablets, Roche, Basel, Switzerland). For the EVs, protease inhibitor was added after purification. Protein concentration was determined using a Qubit Protein Assay Kit (Thermo Fisher Scientific, Waltham, MA, USA) and 30 μg protein was used. For the non-reducing protocol, proteins were prepared in 4× NuPAGE LDS sample buffer (Invitrogen, Waltham, MA, USA). For the reducing protocol, proteins were prepared in 4× NuPAGE LDS sample buffer (Invitrogen, Waltham, MA, USA) and NuPAGE sample reducing agent (Invitrogen, Waltham, MA, USA). The samples were loaded on a 4–12% NuPAGE Novex Bis-Tris gel (Invitrogen, Waltham, MA, USA). Proteins were separated by gel electrophoresis and transferred onto a PVDF membrane. Immunoblotting of CD81 and TSG101 was performed with antibodies against CD81 (sc-166029, Santa Cruz Biotechnology (Dallas, TX, USA); dilution: 1:100) and TSG101 (sc-7964, Santa Cruz Biotechnology; dilution: 1:200). Immunoblotting of Calnexin and RPL22 was performed with antibodies against Calnexin (ab22595, Abcam (Cambridge, UK); dilution: 1:500) and RPL22 (ab229458, Abcam (Cambridge, UK); dilution: 1:500). The HRP-conjugated secondary antibody (DAKO, Agilent (Santa Clara, CA, USA); dilution: 1:5000) was used for band visualization. The membrane was visualized using Pierce ECL Plus (ThermoFisher, Waltham, MA, USA). X-ray films were used to record emitted signals and developed using an Agfa CP1000 developer (Agfa, Mortsel, Belgium).

### 2.7. Small RNA Library Preparation and Sequencing

RNA from EVs (4 × 10^11^) was purified using the miRNeasy Serum/Plasma Advanced Kit (Qiagen, Hilden, Germany) and RNA was eluted in 10 μL RNase-free water. Cellular RNA was purified using the miRNeasy Tissue/Cell Advanced Kit (Qiagen, Hilden, Germany) and RNA was eluted in 30 μL RNase-free water. The quality of purified cellular RNA was determined on a bioanalyzer (Agilent, Santa Clara, CA, USA) using an RNA nano assay (Agilent, Santa Clara, CA, USA).

The Truseq Small RNA Library Preparation Kit (Illumina, San Diego, CA, USA) was used to construct small RNA libraries. One microgram of cellular RNA was used as input (according to the manufacturer’s protocol) and subjected to 12 PCR cycles to amplify the library. For the EV RNA, 5 μL of RNA was used with the following modifications to the protocol: The 3’-adapter, 5’-adapter, and RT primer were diluted 1:2 using RNase-free water and 15 PCR cycles were used to amplify the library. Pippin Prep (Sage Science, Beverly, MA, USA) was used to purify the library. The eluted library was cleaned and concentrated to 10 μL using the MinElute PCR Purification Kit (Qiagen, Hilden, Germany). The Bioanalyzer High sensitivity DNA Analysis Kit (Agilent, Hilden, Germany) was used to determine the size of the library fragments and the KAPA Library Quantification Kit (Roche, Basel, Switzerland) was used to quantify the library. The libraries (from EVs and cells) were pooled and sequenced on a Nextseq 500 sequencing machine (Illumina, San Diego, CA, USA).

### 2.8. Small RNA Sequencing Data Analysis

Raw reads were filtered with FASTX-Toolkit (Version 0.0.13) to trim away low-quality reads and remove adapter sequences to obtain clean reads [34]. The clean reads were mapped to miRBase v22.1 [35,36], allowing zero mismatches. The differential expression of miRNAs was analyzed using the DESeq2 package [37]. miRNAs were deemed significantly de-regulated when the mean of normalized counts ≥100, adjusted *p*-value < 0.1, and log2foldchange |≥1|. The top 400 most abundant miRNAs were used as background miRNA in the following analysis. Gene ontology (GO) enrichment analysis was performed on predicted target mRNAs of miRNAs at biological pathways and cell component levels. To obtain the list of predicted miRNA target genes, we used the R package miRNAtap (v1.16.0) [38] for target prediction, which integrated data from five miRNA databases “DIANA”, “TargetScan”, “miRDB”, “MiRanda”, and “PicTar”. The genes reported by at least three databases as predicted targets of the miRNAs were considered as likely targets. The predicted target mRNAs were analyzed in the R package clusterProfiler (v3.10.1) [39] for the gene ontology enrichment test. Comparing the target genes of de-regulated miRNAs to the target genes of the background miRNAs, annotated ontology terms with an adjusted *p*-value < 0.05 were deemed significantly enriched. The accession number for the sequencing data is PRJNA633988.

### 2.9. Taqman miRNA Assay for EVs

EV RNA was purified using the miRNeasy Serum/Plasma Advanced Kit (Qiagen, Hilden, Germany) according to the manufacturer’s protocol, and RNA was eluted in 10 μL RNase-free water. The RT primers and Taqman probes were ordered from Thermofisher Scientific (miR-182-5p, Assay ID 002599, Catalog number 4427975; miR-183-5p, Assay ID 002269, Catalog number 4427975) (Waltham, MA, USA). cDNA was synthesized using the TaqMan MicroRNA Reverse Transcription Kit (Thermo Fisher Scientific, Waltham, MA, USA) and qPCR was performed using TaqMan Universal Master Mix II (Thermo Fisher Scientific, Waltham, MA, USA). We used the same number of EVs when comparing HR and N EVs from the same batch. Three independent batches of paired HR and N EVs were included, and we used 1.37 × 10^11^ EVs, 1.38 × 10^11^ EVs, and 2.74 × 10^11^ EVs from each batch, respectively. Each EV sample was analyzed in triplicates. The fold change of the miRNA expression was calculated using 2^–ΔCT^ (HR EVs relative to N EVs of the same batch). The *p*-value was calculated using unpaired Student’s *t*-test.

### 2.10. Mass Spectrometry (MS) Sample Preparation

EVs pelleted from C2C12 culture media were washed using PBS and re-pelleted by ultracentrifugation at 100,000× *g* for 2 h at 4 °C (Beckman-Coulter (Brea, CA, USA), Optima L-80-XP ultracentrifuge, type 60Ti rotor) and re-suspended in 100 μL PBS. The protein concentration in the re-suspended EVs was measured using a Qubit Protein Assay Kit according to the manufacturer’s protocol (Thermo Fisher Scientific, Waltham, MA, USA). There were five replicates of EVs for each condition and 50 μg EV protein from each replicate was used for Liquid Chromatography–Tandem Mass Spectrometry (LC-MS/MS) analysis as previously described [40]. Sample preparation, mass spectrometric measurements, and label-free quantification were performed essentially as previously reported [41]. The detailed protocol is included in the Appendix A.

### 2.11. MS Data Analysis

In the data analysis, the proteins detected in all samples were used for differential expression analysis. The fold-change was calculated using the mean counts of the replicates, and the *p*-value was calculated using unpaired Student’s *t*-test. The proteins with log2foldchange |≥2| and *p*-value < 0.05 were considered to be significantly de-regulated. Only the proteins detected in all the replicates of one condition were considered condition-specific proteins. GO enrichment analysis was performed on both significantly de-regulated and condition-specific proteins using the R package clusterProfiler (v3.10.1, Bioconductor) [39] and all the condition-specific proteins and proteins used in the differential expression analysis were used as background proteins in the GO analysis. The annotated ontology terms with adj *p*-value < 0.05 were deemed significantly enriched. The protein–protein interaction (PPI) network was constructed using the R package STRINGdb (v1.22.0, Bioconductor) [42] which provides an interface to the STRING database (http://www.string-db.org (accessed on 20 November 2019)) [43]. Using mouse data (v10) as a reference, we retrieved the interactions between distinct input proteins. Cytoscape (v3.7.1) [44] was used to visualize the PPI network and the app MCODE (v1.5.1) [45] was used to extract the subnetworks (the highly interconnected clusters, which usually imply specific functions or protein families) from the whole PPI network. clusterProfiler was then used to annotate the potential function of the proteins in the subnetworks.

### 2.12. Cell Viability Test

C2C12 cells were seeded in 24-well plates (10,000 cells/well) and cultured at 37 °C with 5% CO_2_/95% air until the confluency reached 30%. The CellTiter-Blue cell viability assay (Promega, Madison, WI, USA) was used to measure cell viability. The working reagent of CellTilter-blue was prepared by adding 1.5 mL CellTilter-blue into 8.5 mL fresh cell media. The media in each well was removed and 400 μL of working reagent was added to each well, and the plate was incubated for 2 h. Then, 150 μL of working reagent was taken out from each well and added into a Costar 96-Well Black Polystyrene Plate (Sigma-Aldrich, St. Louis, MO, USA). The data was collected by a FLUOstar optima microplate reader (BMG Labtech, Ortenberg, Germany). After the first measurements of cell viability, the cells were washed twice in PBS, and then fresh media was added. Thirty micrograms (1 μg/μL) of HR EVs or N EVs were added to the wells, with an equal volume of PBS used as a negative control (three wells/condition). The 24-well plate was first placed into a hypoxia chamber (37 °C with 1% O_2_/5% CO_2_/94% N2) for 5 h and then cultured in a normal incubator (37 °C with 5% CO_2_/95% air). Cell viability was subsequently measured at 24 h, 48 h, and 72 h using the CellTiter-Blue cell viability assay as mentioned above.

### 2.13. Macrophage Activation

RAW 264.7 cells (ATCC, Manassas, VA, USA) were cultured in DMEM supplemented with 10% FBS and 1% P/S at 37 °C with 5% CO_2_/95% air. The cells were seeded in 24-well plates at 50,000 cells/well. Thirty micrograms (1 μg/μL) of purified EVs were added, and an equal volume of PBS was added as a negative control. The plate was cultured for 48 h, the media was then replaced with 800 μL fresh media containing 100 ng/mL lipopolysaccharide (LPS, Chondrex, Woodinville, WA, USA) to induce inflammation, and the cells were cultured for 5 h. Media was removed and 500 μL of Trizol (Invitrogen, Waltham, MA, USA) was added to each well. RNA was purified according to the manufacturer’s protocol, and RNA was re-suspended in 30 μL RNase-free water. Gene expression was quantified using qRT-PCR as described above.

### 2.14. Angiogenesis Assay

HUVECs (ATCC, Manassas, VA, USA) were cultured in T25 flasks (400,000 cells) in endothelial cell basal medium-2 (EBM-2) (Lonza, Basel, Switzerland) supplemented with EGM-2 SingleQuots (Lonza, Basel, Switzerland) at 37 °C with 5% CO_2_/95% air until 70% confluency. Two hundred micrograms (1 μg/μL) of purified EVs were added to HUVEC culture, with an equal volume of PBS as a negative control, and the cells were cultured for 48 h. The Angiogenesis Assay Kit (Abcam, Cambridge, UK) was used according to the manufacturer’s protocol. Fifty microliters of matrigel solution was added to each well of a pre-chilled 96-well plate, and the plate was incubated at 37 °C for 1 h to allow the gel to polymerize. Cells were harvested from the T25 flask, seeded on matrigel (20,000 cells/well), and left to grow for 6 h. The media was removed, and the wells were washed using washing buffer. Staining solution was added to each well and incubated for 30 min at 4 °C. Endothelial tube formation was examined using fluorescence microscopy (green filter Ex/Em = 490/540 nm; Olympus, Tokyo, Japan) and images were analyzed to quantify total tube length and total branching points using the service provided by Wimasis (Onimagin Technologies, Córdoba, Spain).

### 2.15. Transient Middle Cerebral Artery Occlusion Model (tMCAO) and Injection of EVs

The transient middle cerebral artery occlusion (tMCAO) mouse model was used to make a focal cerebral ischemia [46]. C57BL/6 mice (Taconic, Laven, Denmark), 10–12 weeks of age, were anesthetized with ketamine (100 mg/kg) and xylazine (10 mg/kg). Mechanical ventilation was given by oral intubation and the mice were placed on a heating pad with a rectal probe to keep the body temperature at around 37 °C. Cerebral blood flow changes were measured by a laser doppler (Moor Instruments, Millwey, UK) with a probe adhered to the left temporal area, which is supplied by the middle cerebral artery (MCA). The left common carotid artery (CCA) and external carotid artery (ECA) were carefully dissected and ligated, respectively, without affecting the vagus nerve. The left internal carotid artery (ICA) was temporarily closed using a microvasculature clamp. A small hole was cut at the CCA, and a silicone-coated monofilament (602256PK5Re, Doccol, MA, USA) was introduced into the hole and advanced until reaching the origin of the MCA in the circle of Willis. The filament was kept there for 45 min and then withdrawn to allow reperfusion. The decline of cerebral blood flow at the beginning of occlusion and the recovery after reperfusion was confirmed by laser doppler measurements. After surgery, mice were allowed to recover from anesthesia in a 32 °C recovery chamber for 2 h and were then returned to their home cage. Buprenorphine (0.1 mg/kg) was injected subcutaneously during the surgery and three times in the 24 h after reperfusion to reduce the suffering of the mice.

In total, 10 mice were used in this experiment: 3 mice injected with N EVs, 3 mice injected with PBS, and 4 mice injected with HR EVs. EVs were labeled using the ExoGlow-Vivo EV Labeling Kit (Near IR; System Biosciences, Palo Alto, CA, USA) according to the manufacturer’s protocol. The concentration of labeled EVs was measured using NTA. Mice were anesthetized with isoflurane 24 h after the MCAO surgery. Labeled EVs (2 × 10^11^ particles in 250 μL PBS) were injected through the tail vein and 4 h later mice were sacrificed with an overdose of pentobarbital. Brains were removed and sliced into 2 mm sections with a mouse brain matrix. Brain sections were scanned under Odyssey Sa (OSA-0111, 9260, Licor, Bad Homburg, Germany) with a resolution of 20 μm, focus of 3.5 mm, and excitation laser of 785 nm.

### 2.16. Statistical Analysis

The statistical tests used for the small RNA sequencing and MS data analysis are described in the relevant sections above. The other experimental results are presented as the mean ± standard deviations (SD). The unpaired Student’s *t*-test was used to analyze the qRT-PCR data and EV yield data. Ordinary one-way ANOVA with Tukey’s multiple comparison test was used to analyze the data for cell viability, macrophage activation, and angiogenesis. Two-way ANOVA with Sidak’s multiple comparisons test was used to analyze the mouse model data of tMCAO.

## 3. Results

### 3.1. Cyclic Hypoxia-Reoxygenation (HR) Treatment Has No Effect on EV Size and Quantity

In order to evaluate the hypoxia effect on cells, we quantified the expression level of *HIF-1α* in hypoxia-reoxygenation (HR) and normoxic (N) cells by qRT-PCR and found that *HIF-1α* was, as expected, significantly upregulated in HR cells (Appendix A). The average size of HR EVs and N EVs was 123 ± 5 nm and 123 ± 3 nm measured by nanoparticle tracking analysis (NTA) (Figure 1a,b). According to TEM measurements, the EVs appeared slightly smaller with sizes of 70–100 nm and they exhibited a typical “cup-shaped” morphology due to the drying process (Figure 1a,b). According to the ISEV 2018 guidelines [47], we performed a Western blot to characterize the EVs. The Western blot showed that CD81 and TSG101 were enriched in EVs, while Calnexin and RPL22 were enriched in cells (Figure 1c). Based on the EV concentration measured by NTA, and the number of producer cells used, the mean number of EVs secreted per cell was calculated. We found no significant difference between the HR-treated group (3846 ± 785 EVs/cell) and the N group (3691 ± 1098 EVs/cell) (Figure 1d). Therefore, we conclude that cyclic HR treatment does not affect overall EV size and quantity.

### 3.2. HR-Treatment Alters the miRNA Profile of EVs Secreted from C2C12 Cells

To profile the small RNA content of the EVs, small RNAs from N and HR EVs and their respective producer cells were sequenced. We detected 1194 miRNAs in cells and 443 miRNAs in EVs. The overall miRNA content in C2C12 cells decreased upon HR treatment but increased in the secreted EVs (Figure 2a,b). A PCA plot based on the miRNA profiles showed that HR treatment changed the miRNA expression profile in both cells and EVs (Figure 2c,d). The miRNA profiles of the EVs were clearly distinct from their producing cells (Figure 2c), suggesting that the loading of miRNA into EVs is selective rather than the result of passive diffusion.

The differential expression analysis deemed miRNAs to be significantly de-regulated at the mean of normalized counts ≥100, adjusted *p*-value < 0.1, and log2foldchange |≥1|. There were nine significantly up-regulated miRNAs in the EVs upon HR: miR-182-5p, miR-183-5p, miR-25-3p, miR-486-5p, miR-151-3p, miR-30a-3p, miR-191-5p, miR-149-5p, and miR-28a-3p (Figure 3a, Appendix A), while four miRNAs were significantly down-regulated under the same conditions: miR-34c-5p, miR-423-3p, miR-744-5p, and miR-125b-1-3p (Figure 3a, Appendix A). Using the same cutoff value as for the EVs, six miRNAs were significantly up-regulated in C2C12 cells subjected to HR: miR-677-3p, miR-3084-3p, miR-677-5p, miR-3535, miR-5099, and miR-221-5p (Figure 3b, Appendix A), while only miR-92a-1-5p was significantly down-regulated (Figure 3b, Appendix A). In order to validate the RNA sequencing data, the two most deregulated miRNAs (miR-182-5p and miR-183-5p) were chosen for the qPCR. We quantified the expression of miR-182-5p and miR-183-5p using the Taqman miRNA assay in three independent batches of HR EVs and N EVs and found both miRNAs to be significantly upregulated in HR EVs compared to N EVs with *p*-value = 0.007 and *p*-value = 0.015, respectively (Appendix A).

We conducted a GO (gene ontology) analysis on target genes (1659 genes) for the 13 de-regulated miRNAs in HR EVs, using the target genes (77,965 genes) for the top 400 miRNAs in the EVs as background. The analysis found brain development to be the most significantly enriched pathway, while neuronal differentiation and cardiac muscle development were also significantly enriched, in line with the hypothesis that EVs could confer protective function in the brain and heart (Figure 3c). A cell component analysis for the GO terms showed neuron-related components to score high, further suggesting a potentially protective function for the HR EVs in the brain (Figure 3d).

### 3.3. HR Treatment Alters the Protein Profile of EVs Secreted from C2C12 Cells

To investigate protein content in HR and N EVs, we performed mass spectrometry (MS) analysis on five replicates of both HR EVs and N EVs. However, one HR EV replicate was excluded as its ion intensity numbers differed significantly from the other four replicates. The EV markers CD63, CD81, and Tsg101 were detected in HR EVs and N EVs without significantly different levels between HR EVs, and N EVs; however, another known exosome marker, CD9, was not found, in line with a previous report showing that not all EVs express CD9 [48].

Proteins were chosen for further analysis if they were detected in at least all the replicates of one condition. In total, 1611 proteins were identified in the EVs, including 380 up-regulated and 10 down-regulated proteins in the HR EVs. We also found that 230 proteins were HR EV-specific, whereas only 14 proteins were N EV-specific. The EV protein data can be found in the Appendix A.

GO analysis and protein–protein interaction analysis (PPI) were performed on the 390 HR de-regulated proteins, 230 HR EV-specific proteins, and 14 N EV-specific proteins. The 1611 proteins identified in the samples were used as a background for the GO analysis. There were no biological pathways (FDR ≤ 0.05) significantly enriched in the analysis for the 14 N EV-specific proteins. In contrast, GO analysis on the 230 HR EV-specific proteins showed that oxidation–reduction processes, mitochondrial organization, and fatty acid catabolic processes were the most enriched (Figure 4a). In cardiac IRI, mitochondria and cardiomyocytes are damaged due to disruption of the electron transport chain and reduced fatty acid oxidation [49,50]. The enriched pathways for HR EV-specific proteins indicated a possible role for HR EVs in alleviating IRI. Four significant sub-networks appeared in the PPI analysis on HR EV-specific proteins and four de-regulated miRNAs in HR EVs in our data (miR-34c, miR-423-3p, miR-25-3p, and miR-182-5p) were predicted to target these sub-networks (Appendix A).

GO analysis on the 380 up-regulated and 10 down-regulated proteins showed that cellular macromolecule biosynthetic process, organonitrogen compound biosynthetic process, amide biosynthetic process, and regulation of cellular amide were the most enriched (Figure 4b), which indicated a possible role for HR EVs in biosynthesis. The PPI analysis revealed two significant sub-networks that were targeted by three de-regulated miRNAs in HR EVs in our data (miR-25-3p, miR-744-5p, and miR-34c-5p) (Appendix A).

### 3.4. HR EVs Restore Cell Viability of Hypoxia-Treated C2C12 Cells

During ischemia, anaerobic metabolism alters ion exchange and transport and reduces contractile sensitivity, which leads to cell swelling and death [51]. The severity of infarction (cell death) caused by ischemia is generally used to evaluate ischemic damage [11,51]. Restoration of cell viability and proliferation is important to counteract ischemia injury. To test the ability of HR EVs to confer protective signals, we incubated hypoxia-stressed C2C12 cells with EVs from cells grown under HR or N conditions and assayed cell viability under growth. The continuous measurements of cell viability can also indicate the proliferation state of the cells. The results showed that 5 h of hypoxia stress led to reduced viability at 24 h, 48 h, and 72 h compared to normoxic culture with 72 h being significant (adj *p*-value < 0.05) (Figure 5a). Adding EVs purified from both HR and N cultured C2C12 cells, restored cell viability and proliferation of the hypoxia-exposed cells. However, treatment with HR EVs showed a stronger effect on viability (Figure 5a), supporting a more potent effect of HR EVs in reducing ischemic injury.

### 3.5. HR EVs Protect Macrophages from LPS-Induced Inflammation

During reperfusion of ischemic tissue, the immune cells in the returning blood secrete inflammatory substances and the resulting excessive inflammation leads to cell dysfunction and damage [52,53]. RIC was found to suppress the inflammatory response and activate an anti-inflammatory, anti-apoptotic transcription profile to protect damaged tissue [52,54,55,56,57]. In order to test if HR EVs exert a dampening effect on inflammation, we pre-cultured the macrophage cell line RAW 264.7 with HR EVs, N EVs, or PBS (negative control) before treating them with LPS to trigger inflammation. As expected, we found that *IL-1β* expression was strongly induced by LPS in the PBS-treated control macrophages, whereas this effect was significantly lower (adj *p*-value < 0.0001) in macrophages treated with either N or HR EVs (Figure 5b). Moreover, the induction of *IL-1β* was significantly lower (adj *p*-value < 0.05) in HR EV pre-cultured macrophages compared to those pre-cultured with N EVs (Figure 5b). Another cytokine gene (*Tnf-α*) also showed a significantly lower induction (adj *p*-value < 0.05) upon LPS-treatment in RAW264.7 cells pre-cultured with HR EVs (Figure 5b). The lower expression of *Tnf-α* was also shown in N EVs-treated cells, but it was not statistically significant (adj *p*-value = 0.06). Overall, the result suggests that EVs released by skeletal muscle cells can provide a signal to dampen inflammation and that HR treatment could strengthen the function.

### 3.6. HR EVs Enhance Angiogenesis

Angiogenesis is the process of forming new vessels from existing blood vessels, which helps restore oxygen and nutrient supply to the ischemic tissue [58,59,60], and therapeutic angiogenesis is a very important application to protect tissue from ischemic damage or treat ischemic disease [61]. A study showed that RIC reduced cell death in the CA1 region of the brain and promoted angiogenesis in the hippocampus in a chronic cerebral hypoperfusion rat model [62,63]. To test if myoblast EVs released under HR conditions can enhance angiogenesis, we pre-cultured HUVECs with HR EVs and N EVs separately for 48 h and seeded them on matrigel for 6 h before staining. The addition of HR EVs significantly increased the total length of tubes compared to N EV treatment and negative control (PBS) (adj *p*-value < 0.05) and significantly increased the mean of the number of branching points compared to the negative control (PBS) (adj *p*-value < 0.05) (Figure 5c).

### 3.7. HR EVs Accumulate in the Ischemic Hemisphere

It is reported that macrophages are activated in the early stages of inflammation after stroke (around 24 h after stroke onset) which leads to the secretion of inflammatory cytokines that exacerbate brain damage [64]. Our results show that HR EVs protect macrophages from LPS-induced inflammation and we, therefore, examined whether C2C12 EVs can enter the ischemic brain after systemic injection. C2C12 EVs were labeled with a near-infrared dye and injected through the tail vein of mice 24 h after they had been subjected to transient middle cerebral artery occlusion (tMCAO). A silicone-coated monofilament was introduced into the left common carotid artery and kept in the middle cerebral artery for 45min to induce ischemia followed by 24 h reperfusion before EV administration. It was previously found that the EVs could be detected in the brain at 1 h after injection [65]. In a pilot study, we performed intranasal administration (protocol is in Appendix A) and compared the EV signal in the brain taken out at 1 h and 4 h after EV injections. A greater distribution in the brain taken at 4 h after EV administration was clearly shown (Appendix A). However, the amount of injected EVs is better controlled by using tail vein injection, and here, the brain also showed a good EV signal at 4 h after injection (Appendix A). Therefore, the organs were removed and scanned 4 h after EV administration. We noticed that the signal in the liver was much brighter than the other organs and therefore difficult to quantify and compare to the brain (Appendix A). The brains were sectioned and scanned immediately to visualize the distribution of EVs. This showed that both N EVs and HR EVs entered the brain 4h after injection with high fluorescence intensity in the cerebral ventricles. Interestingly, HR EVs showed a higher rate of retention in the brain compared to N EVs (adj *p*-value < 0.01) (Figure 6a,b and Appendix A). Since ischemia was induced in the left hemisphere of all mice, we analyzed the fluorescence intensity in both hemispheres. There was significantly more signal in the ischemic hemisphere compared to the non-ischemic hemisphere in brains from mice injected with HR EVs (adj *p* < 0.05) (Figure 6b,c).

## 4. Discussion

RIC has a well-documented effect on IRI, but the underlying mechanism has remained elusive. A better understanding of the protective effects of ischemic conditioning would be beneficial for patients in the prevention and therapy of IRI. Our hypothesis is that EVs secreted from myoblasts undergoing cyclic hypoxia-reoxygenation treatment may contain protective signals.

We studied the characteristics and functional effects of EVs obtained from myoblast C2C12 cells treated with and without repeated hypoxia-reoxygenation conditions to mimic an in vivo setting. The appearance and numbers of EVs were largely unaltered upon the treatment. This agrees with previous findings from Li et al. who showed that RIC did not increase the number of EVs in rat and human plasma [27], but contrasts the study by Vicencio et al. who reported that RIC increased EV quantity in human plasma [25]. A study from Jeanneteau et al. showed that the number of plasma-circulating EVs pelleted at 21,000 *g* did not increase after RIC in rats and humans, while Annexin V^+^ and endothelium marker positive (CD45 for rat and CD146 for human) EVs were significantly increased [66]. The difference in protocols used for RIC and EV isolation may explain these different results.

Our miRNA sequencing data revealed that the miRNA profile in EVs is very different from their parental cells and that HR treatment changes the miRNA profile in the C2C12 cells and, to an even higher extent, in secreted EVs. We identified miR-182-5p as one of the most abundant and significantly upregulated miRNAs in HR EVs. In hepatocellular carcinoma (HCC), the expression of miR-182-5p is known to be induced by hypoxia, and a miR-182-5p mimic promoted angiogenesis while anti-miR-182-5 inhibited tube formation [67]. This matches our observation, where HR EVs with high levels of miR-182-5p significantly enhanced angiogenesis. MiR-183-5p is expressed in the same cluster as miR-182-5p [68] in compliance with the observation that both miRNAs are significantly up-regulated in HR EVs compared to N EVs. In a study of Parkinson’s Disease, miR-182-5p and miR-183-5p were shown to mediate neuroprotection of dopaminergic (DA) neurons in vitro and in vivo by down-regulating the expression of *FOXO3* and *FOXO1* and enhancing PI3K-Akt signaling [68]. This supports the idea that HR EVs carrying higher levels of miR-182-5p and miR-183-5p could function as neuroprotection. In addition, in our study, MiR-486-5p was abundant and significantly up-regulated in HR EVs. A previous study showed that miR-486-5p was enriched in human endothelial colony-forming cell-derived EVs and helped protect the kidney during an ischemic injury in mice by down-regulating *PTEN* and activating Akt signaling [69]. The GO pathway analysis of EV miRNAs also supports a potential role for HR EVs in tissue protection, especially neurological development and protection, since neuronal differentiation and cardiac muscle development were significantly enriched. Furthermore, neuron-related components were significantly enriched in the cell component of the GO analysis.

HR treatment of C2C12 cells changes the protein content in the secreted EVs and these proteins may also contribute to the protective function of HR EVs as shown in our in vitro assays. The GTP-binding protein Rheb is an HR EV-specific protein found in our data. Rheb belongs to the Ras superfamily of GTP-binding proteins and regulates cell growth and cell cycle progression [70]. These studies support our hypothesis that HR EVs contain protective signals in the form of either miRNAs or proteins. However, in the GO analysis, the enriched biological pathways based on protein and miRNA data were surprisingly different. The difference may be due to different prediction powers from miRNA and protein data, or the different number of candidates included in the analysis. In addition, the difference may indicate that the protein content in EVs might better reflect the metabolic state of the parental cells under HR treatment, and the EV miRNAs and proteins might have different targets in the recipient cells.

In the analysis of RNA-seq data and MS data, we used different cut-off values to select candidates for further analysis, because the RNA-seq and MS analysis are independent datasets, and we would like to identify the most differentially expressed candidates in each dataset. The cut-off value we used for selecting miRNA candidates from the RNA-seq data was adjusted *p*-values < 0.1 and log2foldchange |≥1|. Only two up-regulated miRNAs and one down-regulated miRNA were found when using a cut-off of *p*-values < 0.05 and log2foldchange |≥2| (as used for the MS data). Therefore, to include more miRNA candidates in the pathway analysis, we used log2foldchange |≥1| as a cut-off.

It has been shown that protective signals for RIC can be transmitted across the blood–brain barrier (BBB) and reduce stroke injury [71,72]. In our study, HR EVs and N EVs accumulated in brains when administered to mice with stroke and with a slightly higher accumulation in the ischemic part. It is known that stroke damages the BBB [73] and this could increase EV entry. Chen et al. found that EVs derived from HEK293T cells could cross the BBB model formed by brain microvascular endothelial cells upon treatment with TNF-α to mimic stroke-like conditions [74]. However, we found that HR EVs had higher accumulation in the brain than N EVs, which may indicate the potentially protective function of HR EVs. However, further experiments are needed to investigate the possible functional role HR EVs play in the protection of the brain in stroke injury. The strong anti-inflammatory response observed in macrophage cytokine signaling in our study suggests that EVs may act in vivo primarily by dampening the activated macrophages to avoid self-destruction of tissue. Hence, to study the full functional capacity of EVs in the future, fully immunocompetent in vivo systems are needed.

Regarding the amount of EVs used in different in vitro and in vivo experiments, both particle number and protein amount are used widely in the literature [75,76,77,78]. We measured the particle number and protein concentration of our EV samples and found that HR EVs had slightly a higher protein concentration with the same particle number. Therefore, in order to add a similar level of active molecules, we chose a protein concentration for the in vitro studies. The in vivo experiment quantifies EV entry in the brain, and we, therefore, chose to inject similar particle numbers. The amount of EVs used in the in vitro assays or in vivo experiments varies a lot among different studies [78,79,80] and reflects the different experimental reaction volumes and setups. We used the same protein amount of EV in each well of 24-well plates of the cell viability assay and macrophage activation assay. In the cell viability study, the media containing EVs was removed after 24 h (compared to the 48 h in the macrophage experiment), and we, therefore, increased the EV amount added. Increasing the amount of EVs could potentially increase the protective effect, but this was not tested in the present study.

Our results clearly show a change in EV composition in response to the cyclic hypoxia-reoxygenation and a positive effect of EVs on cell proliferation in vitro. This highlights that myoblast EVs could be part of the mechanism to deliver protective signals generated by RIC in distant limbs to targeted tissues, and, as such, our findings open the possibility of applying HR-treated EVs for therapy. However, our study is based on an in vitro mimic of RIC (HR), and future in vivo studies are required to understand the full complexity of the remote protective effects conferred by RIC.

## Figures and Tables

**Figure 1 biomedicines-09-01211-f001:**
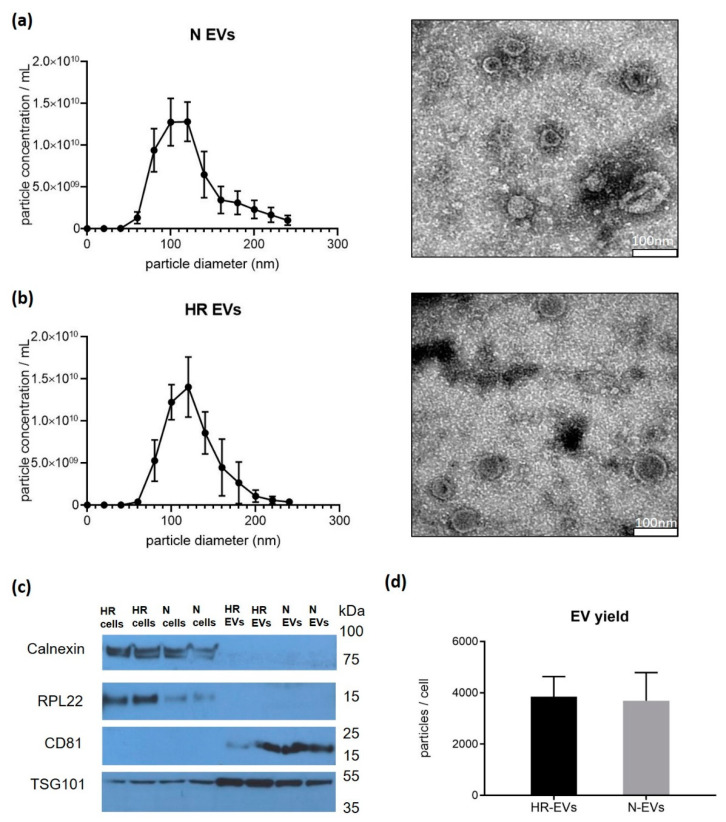
Characterization of EVs secreted from normoxic cultured (N) and cyclic hypoxia-reoxygenation treated (HR) C2C12 cells. Size distribution using nanoparticle tracking analysis (NTA) and transmission electron microscopy (TEM) of (**a**) N EVs and (**b**) HR EVs. NTA data is presented as the mean ± SD (*n* = 5). Scale bar of EM image: 100 nm. (**c**) Characterization of Calnexin, RPL22, CD81, and TSG101 using Western blot. Full-length blot is presented in Appendix A. (**d**) Yield comparison of HR EVs and N EVs. Data is presented as the mean ± SD (*n* = 3).

**Figure 2 biomedicines-09-01211-f002:**
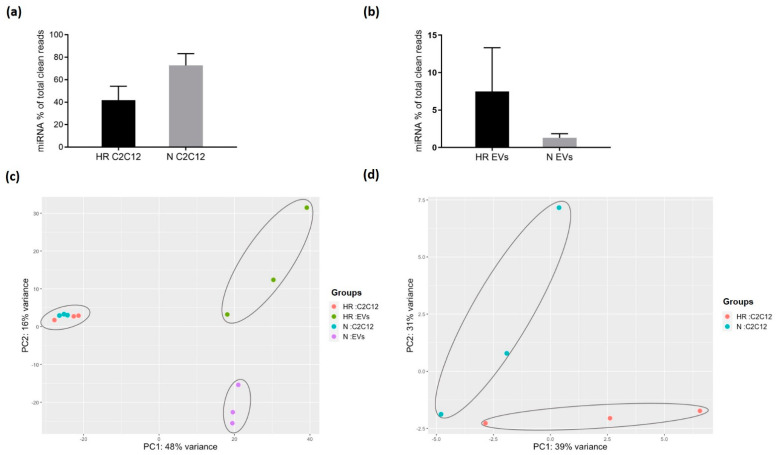
miRNA data analysis. Mapped miRNA reads as a percentage of total clean reads in (**a**) C2C12 cells and (**b**) C2C12 EVs. Data is presented as the mean ± SD (*n* = 3). Principal component analysis (PCA) of miRNA profile in (**c**) C2C12 cells and EVs, and (**d**) C2C12 cells. N: normoxic cultured; HR: cyclic hypoxia-reoxygenation treated.

**Figure 3 biomedicines-09-01211-f003:**
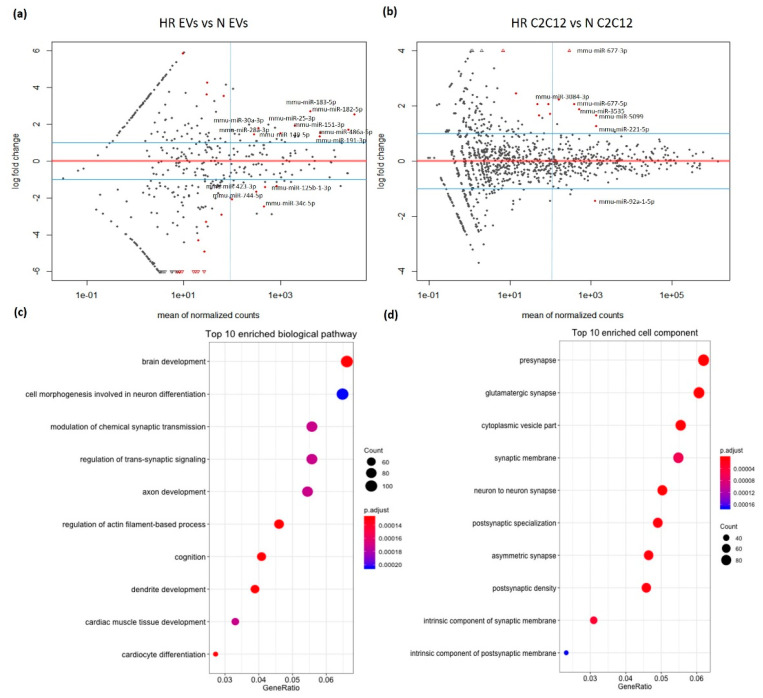
Differential expression analysis of miRNAs in EVs and C2C12 cells under normoxic culture (N) and cyclic hypoxia-reoxygenation treatment (HR). Differential expression analysis of miRNAs using DESeq2 in (**a**) EVs and (**b**) C2C12 cells. Red dots: adj *p*-value < 0.1. Dots with miRNA names: normalized counts ≥ 100, adj *p*-value < 0.1, and log2foldchange |≥1|. (**c**) The top 10 biological pathways based on significance and (**d**) the top 10 cell component-enriched terms based on significance from the Gene Ontology (GO) enrichment analysis on differentially expressed miRNAs in EVs produced from C2C12 cells undergoing HR treatment. The point size is proportional to the number of genes in the GO term. The color shows the adjusted *p*-value.

**Figure 4 biomedicines-09-01211-f004:**
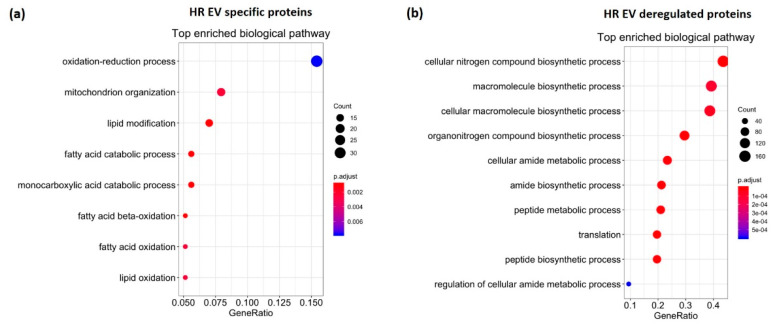
Gene ontology (GO) analysis of EV proteins. The most enriched biological pathways based on the significance of (**a**) specific proteins in EVs secreted from hypoxia-reoxygenation (HR)-treated C2C12 cells and (**b**) differentially expressed proteins in HR EVs compared to normoxic-cultured (N) EVs.

**Figure 5 biomedicines-09-01211-f005:**
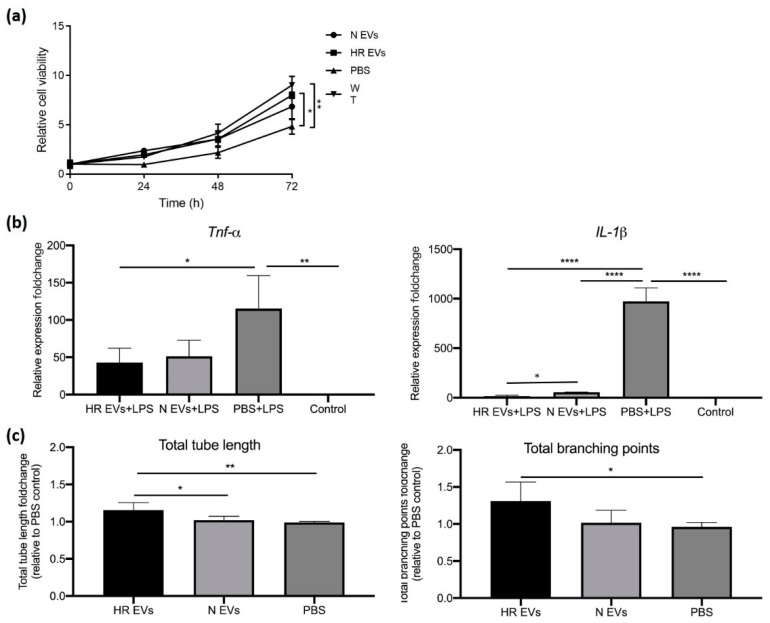
Analysis of HR EV protective function in vitro. (**a**) Evaluation of cell viability of C2C12 cells co-cultured with N EVs, HR EVs, or PBS. WT: C2C12 cells cultured under normoxic conditions. The data is presented as the mean ± SD; *n* = 3 independent experiments; one-way ANOVA with Tukey’s multiple comparison test, * adj *p*-value ≤ 0.05, ** adj *p*-value ≤ 0.01. (**b**) Quantification of cytokine mRNA expression in EV co-cultured RAW 264.7 cells after LPS induction (qPCR). *GADPH* was used to normalize the data and the fold-change is relative to control. The data is presented as the mean ± SD; *n* = 3 independent experiments; one-way ANOVA with Tukey’s multiple comparison test, * adj *p*-value ≤ 0.05, ** adj *p*-value ≤ 0.01, **** adj *p*-value ≤ 0.0001. (**c**) Quantification of total branching points and total tube length of in vitro angiogenesis images using a service provided by Wimasis. The data is showed as fold-change relative to PBS control. The data is presented as the mean ± SD; *n* = 3 independent experiments; one-way ANOVA with Tukey’s multiple comparison test, * adj *p*-value ≤ 0.05, ** adj *p*-value ≤ 0.01.

**Figure 6 biomedicines-09-01211-f006:**
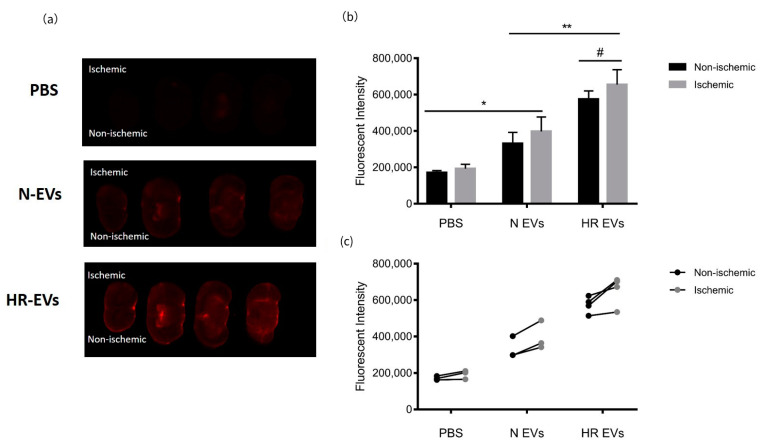
Distribution of myoblast EVs in ischemic brains. (**a**) Fluorescent images of brain sections from stroke mice injected with PBS, N EVs, and HR EVs, respectively. Ischemic hemispheres are the top hemispheres (left-side) for all the images. (**b**) Quantification of fluorescent intensities in both hemispheres by Fiji ImageJ. The data is presented as the mean ± SD; *n* = 3 in PBS and N EVs group, *n* = 4 in HR EV group. Two-way ANOVA with multiple comparisons, * adj *p*-value < 0.05, ** adj *p*-value < 0.01, # adj *p*-value < 0.05 (non-ischemic vs ischemic in the HR EV-treated group). (**c**) Individual values of fluorescent intensities in both hemispheres.

## Data Availability

The accession number for the sequencing data is PRJNA633988. The data of Mass spectrometry is in the Appendix A. The datasets generated during and/or analyzed during the current study are available from the corresponding author on reasonable request.

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
