# Peer review of "Cyclic Hypoxia Conditioning Alters the Content of Myoblast-Derived Extracellular Vesicles and Enhances Their Cell-Protective Functions"

_biomedicines, 2021, doi:10.3390/biomedicines9091211_

Round 1

Reviewer 1 Report

The manuscript by Yan and collaborators investigates EVs derived from murine myoblasts subjected to cyclic hypoxia conditioning, focusing on their potential beneficial effect on ischemia-reperfusion injury. The authors performed EV characterization and some functional experiments to test EV effects. However, there are several aspects of their work that should be further explained and/or justified. I consider that this work could the interesting for the readers of Biomedicines journal, but the following points should be addressed:

Major comments

  1. There are some aspects of the experimental procedures that should be clarified and further explained. It should be clarified in lines 110-112 whether the N dishes (normoxic conditions) were also kept for 24h more before EV collection. Regarding qRT-PCR, 2.2 and 2.9 subsections are presented. It should be explained whether 2.2 refers only to the experiments carried out in cells (as 2.9 refers to EVs). In both sections authors should state whether samples were amplified in triplicates, and in section 2.9, the references of primers from Thermofisher should be added.
  2. Authors used the same amount of RNA for the analysis of cells, but particle number was used for EVs. I consider that the use of particle number instead of RNA concentration should be explained. In addition, bioanalyzer was used only for cellular RNA, why were EVs not evaluated? In section 2.9 the amount (or particle number) used should be stated. Regarding RNAseq analysis, I think that authors should have written “log2foldchange |≥1|”, as they did not only select the upregulated miRNAs, but also the downregulated ones.
  3. The primary and secondary antibody dilutions used for western blot analysis have to be specified.
  4. I have many doubts about the experiments performed to evaluate cell viability with CellTiter-blue. Manufacturer’s protocol indicated that 96 or 384 well plates could be used. But here, 24 well plated were used. Which medium and reagent volumes were used? Manufacturers also say that incubation time should be tested (normally 1-4h). What was the incubation time here? And more important, they say that it is designed as an endpoint assay. Authors indicated that “after the first measurements of cell viability, cells were washed twice in PBS and then fresh media was added”. So, CellTiter Blue was used 4 times in the same cells? Did authors measure fluorescence with or without the well plate lid? Did they consider the toxicity of the reagent? (See https://www.ncbi.nlm.nih.gov/books/NBK144065/)
  5. For the macrophage activation and angiogenesis experiments, first EVs were added and then cells were challenged, which could be useful to evaluate their potential protective function. In contrast, tMCAO was performed first and then EVs were injected to the mice. I think that the rationale for this procedure should be explained. Besides, for the functional assays EVs to be used normalized by concentration, and not by particle number. Why? And how was the concentration measured? The protein concentration measurement is not presented. In addition, different EV treatments were used: for cell viability 30mg/10,000 cell, for macrophages 30mg/50,000 cell for 48h, for angiogenesis 200mg with cell number not stated and for 24h, and finally for tMCAO 2e+11 particles. Authors should discuss on these differences.
  6. The statistical analyses performed should be clarified and justified. Why was ANOVA used when comparing only two groups? How did they get the individual p-values and adjusted p-valued between two groups? ANOVA does not give these data (for instance for macrophage activation). Why did authors use paired t-test for the miRNA validation analysis? Even if from the same batch they are not the same samples. They did not use this test to compare HR and N samples in other cases.
  7. The intranasal administration of EVs is only commented in the results section and the figure presented in the supplementary, but it was not explained in the methods. All the information regarding this procedure should be added, how it was done, how many EVs were administered…
  8. The comparisons, statistic tests and obtained results for the in vivo experiments need to be better presented. The significant differences mentioned in the text (lines 486-491) are not supported by the graphs presented in Figure 6. Similarly, the legend of the figure is not in line with the graphs.
  9. After the promising results obtained in the in vitro experiments, I was expecting the authors to check the effect of EVs in the in vivo model. It would be interesting to evaluate if macrophage infiltration/activation and other brain damage markers are reduced with EVs, or whether EV administration enhances protective signals. For instance, immunostaining could be performed in brain samples obtained from the tMCAO animals.

Minor comments

  1. In my opinion, the title of the manuscript is vague. More precise information than “protective signals” could be given based on the results obtained.
  2. When stating the author contributions there are initials that do not correspond to any of the 15 authors, at list as they are spelled in the author list: TTG, JYS, IJL, AET and KRK.
  3. In the abstract, “N EVs” are mentioned, but this abbreviation has still not been presented.
  4. I consider that the term EV should be used all times. Authors comment about exosomes size and formation as well as some of the characteristic markers, but as they did not prove whether the particles they isolated and study are exosomes or other type of EV, I think that it could be more appropriated to use always the term EV and also take them as a reference. Besides, the nomenclature based on size it is not recommended by MISEV2018, so authors could reconsider the sentence in line 47. As authors explain in section 2.3 that EVs were passed through a 0.22mm filter, so they could mention that they work with “small EVs”.
  5. Regarding EVs, in my view, the sentence in lines 52-53 does not show all the work that has already been carried out. Maybe the sentence could be modified and a publication that summarizes current knowledge cited (for instance the one from van Niel et al. (2018) Shedding light on the cell biology of extracellular vesicles). This reference could also be used instead of the refence number 19, which is just a comment.
  6. In the introduction, it would be interesting to mention the animal model or system used in the experiments by Li et al. (ref. 27), as it is done for the other studies that are mentioned. Similarly, when the previous work by the authors is commented (ref. 28) it should be mentioned that it was carried out with humans. Finally, it is mentioned that Cai et al. (ref. 30) used C2C12 cells and hypoxia-oxygenation, but the main results or output of this work is not commented.
  7. Please check the sentence in lines 76-78, it does not make sense. I think that the information in final sentence of the introduction could also be presented in a clearer manner.
  8. APC abbreviation should be presented, line 80.
  9. O2 and CO2, as well as 2-DCT subscript and superscripts should be corrected throughout the text. The use of qRT-PCR or RT-PCR use should also be unified.
  10. I think that it should be discussed why different cut-offs (fold change and p-values) were used for RNAseq and MS data analysis.
  11. I consider that authors could present a supplementary file with the list of target genes used for the GO analysis. Or at least, the number of target genes in the deregulated group and in the background should be mentioned.
  12. The significant enrichment of miRNAs with target genes related to neural pathways and cell components in HR EVs should de discussed in the manuscript. Moreover, the enrichment at the miRNA level but not at the protein level could be discussed.
  13. In Figure 3, (a) and (b), similar to the lines shown for fold change, it would be useful to show a line for the cut-off of normalized counts. In (d) the complete names of the last two cell components cannot be read. And in the legend, it is said that dot with miRNA names have a p-value < 0.05, but it should be 0.1, as indicated in the text ad shown in Table S2. What about miR-22-3p? It is shown in figure 3(a), but it is not included in the text and table as a significantly deregulated miRNA.
  14. Authors only validated two of the deregulated miRNAs found by RNAseq. Why? I consider that a short explanation should be included in the manuscript.
  15. In lines 381-382 authors state that “exosome” markers were detected at comparable levels in HR and N EVs, but when checked in the excel file, we see that CD81 and CD82 have significant differences between the two groups. This should be commented by the authors.
  16. I think that the sentence in lines 405-406 could be presented before, maybe in line 387, so that readers know that the file is available and can check the mentioned proteins.
  17. The information presented in lines 418-420 is not shown in the figures. Figure 5a only shows the relative viability, so we cannot see the differences obtained with the 5h under hypoxia. In addition, I consider that a couple of sentences explaining that the experiment aims to measure cell proliferation could be added in the results section. This is only commented in the last paragraph of the discussion, and no link to the viability assay is mentioned.
  18. In lines 437-439 the significant effect of HR EVs is mentioned, but it should be stated that the effect of N EVs is nearly the same.

Reviewer 2 Report

The present research article is the result of well-designed, well-executed research work. The authors demonstrated their hypothesis with experimental results. However, I have the following suggestions:

  1. Please include a graphical abstract.
  2. Please write O2, CO2 instead of O2, CO2.
  3. Include a space before units.
  4. Please provide large clear EM images in Figure 1 a,b for the clear demonstration.
  5. Please improve the text in Figure 2c, d.
  6. Enlarge the figures to visualize all the texts in Figure 3.
  7. Use Italic fonts wherever required (for example, in vitro, in vivo, etc.)

Round 2

Reviewer 1 Report

Authors addressed most of the comments, and their work is now more precisely presented. However, I consider that the following points should still be clarified:

1. I would like to thank the authors for specifying the EV concentration or particle number selection. I consider that this text, or an adapted version of it, should be included in the discussion section of the manuscript.

“Both particle number and protein amount are used widely in the literature. We measured particle number and protein concentration of our EV samples and found that HR EVs had slightly higher protein concentration with the same particle number. Therefore, in order to add a similar level of active molecules, we chose protein concentration for the in vitro studies. The in vivo experiment quantifies EV entry in the brain, and we therefore chose to inject similar particle numbers.

The amount of EVs used in the in vitro assays or in vivo experiments varies a lot among different studies, and reflects the different experimental reaction volumes and setups. We used the same protein amount of EV in each well of 24 well plate of cell viability assay and macrophage activation assay. In the cell viability study, the media containing EVs was removed after 24 h (compared to the 48 h in the macrophage experiment), and we therefore increased the EV amount added."

Besides, authors could discuss the limitations of their study, as they did an arbitrary selection of EV concentrations to be added, and different treatments could result in different outcomes.

2. Similarly, authors’ comments about the selection of cut-off values could be included in the manuscript:

"The RNA-seq and MS analysis are independent data sets and we would like to identify the most differentially expressed candidates in each dataset. The cut-off value we used for selecting miRNA candidates from the RNA-seq data was adjust p-values <0.1 and log2foldchange |≥1|. Only two up-regulated miRNAs and one down-regulated miRNAs were found when using a cut-off of adjust p-values <0.1 and log2foldchange |≥2| (as used for the MS data). To include more miRNA candidates in the pathway analysis, we therefore used log2foldchange |≥1| as cut-off. For the MS data, we found 380 upregulated proteins in HR EVs and 10 down-regulated ones with the cut-off log2foldchange |≥2| and p-values <0.05."

Please note that the sentence “using a cut-off of adjust p-values <0.1 and log2foldchange |≥2| (as used for the MS data)” does not match the p-value <0.05 indicated in the manuscript for the MS analysis.

3. I think that my previous comment about the GO analysis of target genes was not properly understood by the authors. I was asking about the analysis of miRNA target genes, but their reply was about the GO analysis based on protein data. Going back to my first comment, I think that authors should present a supplementary file with the list of predicted miRNA target genes used for the GO analysis. In the current version of the manuscript, we only know the 13 miRNAs that were found de-regulated and that 400 miRNAs were used as background, but the IDs of these 400 miRNAs and the number and IDs of predicted target genes that were identified and used for the GO analysis based on miRNA data are not indicated.

4. I would like to thank authors for including a couple of sentences discussing the different results of GO based on protein and miRNA data. Apart from the prediction power and the different number of candidates included in the analysis, the observed differences could be related to the actual EV content, no? That the miRNAs and proteins secreted in EVs have different targets. Besides, my previous comment about the significant enrichment of miRNAs with target genes related to neural pathways and cell components in HR EVs is still not discussed. Could authors hypothesize whether the enrichment neural components could be linked to potential neuroprotective signals?
